# Diagnostic Challenge of Localized Tenosynovial Giant Cell Tumor in Children

**DOI:** 10.3390/diagnostics15030281

**Published:** 2025-01-24

**Authors:** Jiro Ichikawa, Satoshi Ochiai, Tomonori Kawasaki, Kojiro Onohara, Masanori Wako, Hirotaka Haro, Tetsuo Hagino

**Affiliations:** 1Department of Orthopaedic Surgery, Interdisciplinary Graduate School of Medicine, University of Yamanashi, Chuo 409-3898, Yamanashi, Japan; wako@yamanashi.ac.jp (M.W.); haro@yamanashi.ac.jp (H.H.); 2Department of Orthopaedic Surgery, National Hospital Organization (NHO), Kofu National Hospital, Kofu 400-8533, Yamanashi, Japan; hxcmk230@ybb.ne.jp (S.O.); tmhagino@amber.plala.or.jp (T.H.); 3Department of Pathology, Saitama Medical University International Medical Center, Hidaka 350-1298, Saitama, Japan; tomo.kawasaki.14@gmail.com; 4Department of Radiology, Interdisciplinary Graduate School of Medicine, University of Yamanashi, Chuo 409-3898, Yamanashi, Japan; konohara@yamanashi.ac.jp

**Keywords:** tenosynovial giant cell tumor, localized, differential diagnosis, knee pain in children, magnetic resonance imaging, histopathology, treatment

## Abstract

This report describes a rare case of a pediatric tenosynovial giant cell tumor (TSGCT) with a delayed diagnosis. A 9-year-old boy presented with a 3-month history of knee pain and swelling, initially attributed to a femoral non-ossifying fibroma and arthritis based on computed tomography findings and slightly elevated C-reactive protein levels. The symptoms persisted despite medical treatment. Magnetic resonance imaging (MRI) revealed a tumor in the posterior compartment. He underwent surgery, and the pathology confirmed the diagnosis of localized TSGCT. Six months postoperatively, the patient remained asymptomatic. Pediatric knee pain is a complex symptom associated with inflammatory conditions and benign and malignant tumors. Benign tumors, as in this case, can be misdiagnosed as arthritis, delaying diagnosis and treatment. MRI is recommended in cases involving symptom persistence. However, histopathological, immunohistochemical, and morphological examinations are crucial for definitive diagnosis, particularly when the imaging findings are inconclusive.

##  


Figure 1A 9-year-old boy presented with a 3-month history of left knee pain and effusion. Two months before presentation, his family consulted with a local physician. An X-ray revealed a radiolucent lesion with a sclerotic rim on the medial aspect of the distal femoral diaphysis (**A**; red arrow). However, no periosteal reaction or intraosseous mass was observed (**A**,**B**). The final diagnosis was benign bone tumor and growing pains, and acetaminophen was prescribed. The knee pain improved but then worsened, followed by knee effusion for 2 weeks. Considering the potential progression of the bone tumor, he was referred to our hospital for further examinations. A physical examination revealed no fever and the presence of knee joint pain and swelling, with a good range of motion. The results of laboratory tests showed a white blood cell (WBC) count of 6770/µL (normal: 3300–8600) and C-reactive protein (CRP) of 0.24 mg/dL (normal: 0–0.14), suggesting mild inflammation. A rheumatoid factor and antinuclear antibodies analysis showed negative results. The patient played tennis but had no history of trauma.
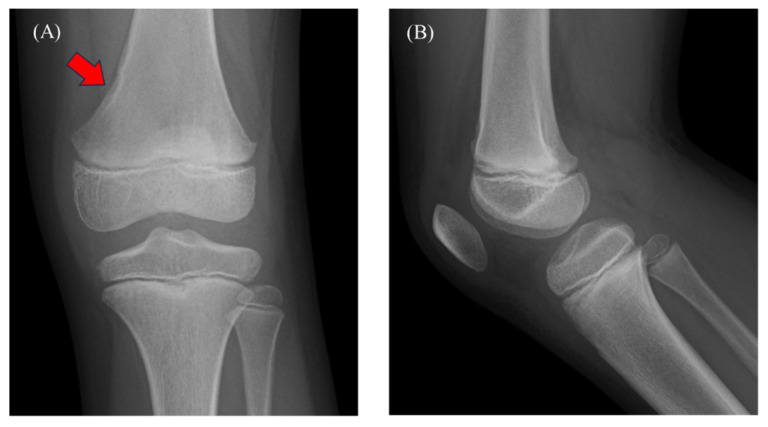

Figure 2Computed tomography (CT) showed a tumor-like lesion on the medial aspect of the distal femoral diaphysis, suggesting a non-ossifying fibroma (NOF) (**A**), along with joint effusion (**B**,**C**). No soft tissue tumor was found. The initial diagnosis was NOF and arthritis or growing pains, and acetaminophen was administered for 30 d. The knee pain and effusion persisted despite continued management, and magnetic resonance imaging (MRI) was performed. Growing pains are common in children; nonetheless, the differential diagnosis of mechanical, infectious, inflammatory, and neoplastic conditions is challenging [1]. Considering the patient’s age, primary malignant bone tumors, such as osteosarcoma and Ewing’s sarcoma, and other malignancies, including leukemia and neuroblastoma, were suspected, underscoring the need for a definitive diagnosis [1]. The presence of knee pain, joint effusion, and mildly elevated CRP levels raised the possibility of a serious underlying condition. After excluding NOF as the cause of these symptoms, we suspected juvenile idiopathic arthritis (JIA) and initiated acetaminophen treatment and follow-up. JIA is a chronic inflammatory disorder that typically affects children < 16 years and is characterized by arthritic symptoms lasting for at least 6 weeks [2]. The clinical manifestations of JIA are broad, with monoarticular, polyarticular, or systemic onset. Diagnosis is challenging and is made by exclusion because of the lack of a definitive diagnostic test [3]. Although monoarticular JIA was suspected, the diagnosis of other conditions, including septic arthritis, reactive arthritis, malignant tumors, hemophilia, and trauma, needed to be excluded [3]. The mean WBC count and CRP level in patients with JIA are 8810/µL and 4.3 mg/dL, respectively. Although these values were higher than in our patient, the wide range of CRP values in JIA does not exclude diagnosis in patients with lower values [4]. Furthermore, although pain and swelling are typical symptoms of tenosynovial giant cell tumor (TSGCT), the absence of tumor lesions on the imaging and the patient’s age excluded TSGCT diagnosis. MRI should have been performed because of persistent knee swelling and pain. A study involving nine children with knee TSGCT reported a mean diagnostic delay of 18 months (range: 3–48 months), and 44% of cases were misdiagnosed as JIA [5]. The diagnosis of septic arthritis must be ruled out in cases of joint effusion. However, the absence of fever and inflammatory markers precluded the need to perform joint aspiration. Joint fluid culture and analysis can help confirm the diagnosis.
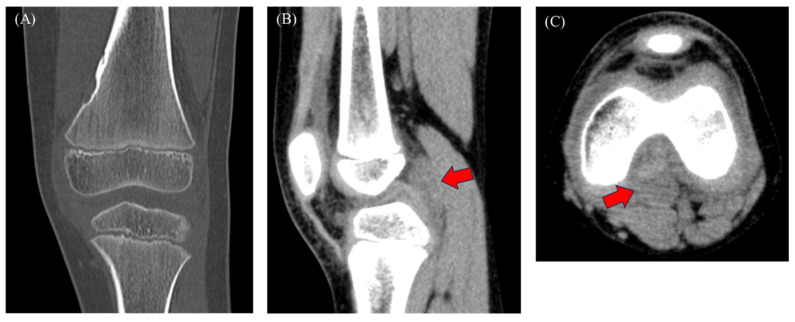

Figure 3MRI showing a hypointense mass on T1-weighted images (**A**; red arrow), T2-weighted images (**B**), and short-TI inversion recovery images (**C**; red arrow), with hypointensity on T2*-weighted images (**D**), indicating blooming. The tumor was located behind the posterior cruciate ligament, and joint effusion was present. Although rare in children, TSGCT was suspected because of its intra-articular location and the presence of hemosiderin deposition. Other possible conditions were synovial hemangioma, hemophilic arthropathy, expanding hematoma, and chronic synovitis. CT scans revealed a mass with a density that was similar to that of muscle (Figure 2B,C; red arrow). MRI is useful for detecting intra-articular masses. Characteristic findings of TSGCT include isointensity or hypointensity on T1- and T2-weighted images with variable degrees of contrast enhancement [6]. Similar MRI findings have been observed in children with TSGCT [7]. Blooming, a common feature of TSGCT, is common in diffuse TSGCT (DTSGCT) and rare in localized TSGCT (LTSGCT) [6]. CT imaging may have limitations in detecting TSGCT without bone invasion, as demonstrated by the initial imaging in this case. Hemosiderosis synovitis (HS) is an important differential diagnosis for TSGCT. A few studies identified characteristics distinguishing HS from TSGCT, including lateral meniscus and cartilage injury, synovial thickening, and contrast enhancement. Nonetheless, the number of reported cases in children is small [8]. In some cases, MRI failed to detect TSGCT in the preoperative diagnosis of meniscal tears [9], and TSGCT is difficult to differentiate from septic arthritis [10], indicating the potential limitations of MRI.
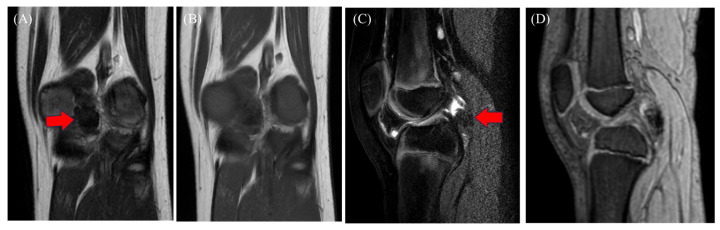

Figure 4Arthroscopic resection of the tumor was performed to establish a definitive diagnosis. The macroscopic examination revealed a brown mass (**A**). Microscopy showed histiocyte-like mononuclear cells, fibroblast-like cells, and osteoclast-type multinucleated giant cells with focal areas of foam cells (**B**,**C**). Prussian blue staining showed hemosiderin-laden macrophages (basophilic cells) (**D**), and immunohistochemistry (IHC) revealed the presence of histiocyte-like mononuclear cells that were positive for clusterin (**E**) and CSF-1 (**F**) and negative results for desmin (**G**). Additionally, the osteoclast-like multinucleated giant cells were positive for CD68 (PGM-1) and negative for CD163. Therefore, the final diagnosis was TSGCT, which includes tumors arising from synovial joints, bursae, and tendon sheaths with synovial differentiation and pigmented villonodular synovitis. LTSGCT is more common than DTSGCT. The hands, particularly the fingers, are the most common site of LTSGCT, accounting for approximately 85% of cases [11]. Although TSGCT affects predominantly women [11], Du et al. [12] reported that the incidence of pediatric LTSGCT was higher in boys, with an equal distribution between hands and feet. LTSGCT usually occurs in a single joint; however, a few cases of bilateral involvement have been reported [11,12]. LTSGCT affects the following knee structures in decreasing order: lateral recessus, suprapatellar recessus, Hoffa fat pad, posterior compartment, femoral notch, and lateral and medial compartments [13]. The typical age of onset is 30–50 years, and pediatric cases are rare [5,11]. Differential diagnoses for TSGCT include HS and giant cell tumor of soft tissue (GCTST) [14,15]. GCTST shares histological similarities with giant cell tumors of bone, comprising mononuclear cells and osteoclast-like multinucleated giant cells, sometimes with metaplastic bone formation [16]. Additionally, the key features that differentiate TSGCT from GCTST include the heterogeneous distribution of multinucleated giant cells, the small number of giant cell nuclei, the presence of large histiocyte-like mononuclear cells, stromal collagenization, and a heterogeneous population of foamy macrophages and inflammatory cells [15]. However, HS is difficult to differentiate from TSGCT. Unlike TSGCT, HS lacks clusters of oval histiocyte-like mononuclear cells and has fewer osteoclastic multinucleated giant and foam cells [14]. Clusterin, CSF-1, and desmin are common IHC markers of TSGCT [11]. Clusterin was shown to be highly expressed in large mononuclear cells in both DTSGCT and LTSGCT [17]. Moreover, 77%, 75%, and 80% of cases of TSGCT, DTSGCT, and LTSGCT, respectively, were positive for CSF-1 [18]. Nevertheless, the utility of these markers in differentiating GCTST and HS from TSGCT is unknown. Desmin, with a positivity rate of 45–80% in TSGCT, is used to identify dendritic processes; however, it lacks specificity because of its expression in various tumors [11]. The primary treatment for TSGCT is arthroscopic or open resection. The rate of recurrence is significantly lower in LTSGCT than in DTSGCT. However, the recurrence rates are similar between open and arthroscopic resection in LTSGCT; thus, the latter is the preferred approach [13]. Conversely, the recurrence rates between open and arthroscopic resection in DTSGCT are unknown [19]. Regarding other treatment modalities, radiation therapy is recommended for TSGCT because of its benign nature and potential long-term adverse effects [19]. The CSF-1R inhibitor vimseltinib treats TSGCT by targeting CSF-1. In this respect, biologics can become the gold standard for treating primary or recurrent TSGCT in children, potentially replacing surgery. A phase 3 trial involving patients aged ≥18 years demonstrated that vimseltinib was more effective for tumor suppression, clinical function, and symptom improvement than placebo [20]. Nonetheless, the effectiveness of vimseltinib in children is unknown. In conclusion, this report describes a rare case of pediatric LTSGCT of the knee with a delayed diagnosis. Arthritis, particularly JIA, is an important differential for pediatric knee pain. Similarly, other conditions, including malignant bone tumors and trauma, must be eliminated. In cases of persistent knee pain and swelling, MRI is essential for detecting tumors and evaluating the degree of arthritis. However, even with MRI, a definitive diagnosis involves a histopathological examination.
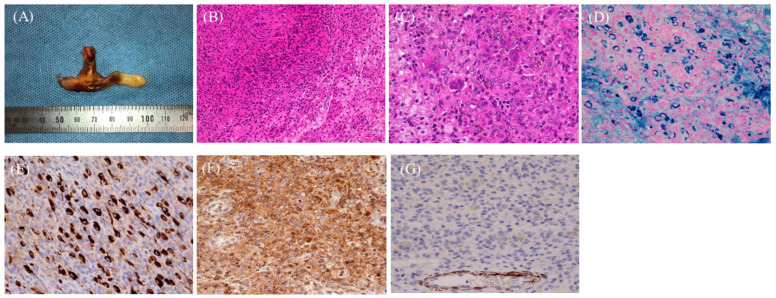



## Data Availability

The data presented in this study are available from the corresponding author upon reasonable request.

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
