# Peer review of "Diagnostic Challenge of Localized Tenosynovial Giant Cell Tumor in Children"

_diagnostics, 2025, doi:10.3390/diagnostics15030281_

Round 1

Reviewer 1 Report

Comments and Suggestions for Authors

The images presented here have significant clinical value and offer a critical reflection on the diagnostic challenges of TCGTS. The images included are of great value, however, authors are advised to start the work with a brief theoretical foundation and then develop the content presented. It would be helpful to include a concise and precise physiological contextualization of the pathology.

In addition, authors are advised to carefully review the image numbering and ensure that each subimage is accurately cited in the text. This aspect is essential to ensure a complete and accurate understanding of the work.

Author Response

Reviewer Comments:

Dear Reviewers,

Thank you for your thoughtful feedback and comments on our manuscript. We appreciate your efforts and the time you provided to improve the manuscript. Your insights have enhanced the quality of our study. Our point-by-point responses to your comments are shown in red font below. The changes to the manuscript are shown in red font.

Reviewer 1

The images presented here have significant clinical value and offer a critical reflection on the diagnostic challenges of TCGTS. The images included are of great value, however, authors are advised to start the work with a brief theoretical foundation and then develop the content presented. It would be helpful to include a concise and precise physiological contextualization of the pathology.

In addition, authors are advised to carefully review the image numbering and ensure that each subimage is accurately cited in the text. This aspect is essential to ensure a complete and accurate understanding of the work.

Response: Thank you for your positive feedback and comments on our manuscript. Image numbering was formatted as requested.

Reviewer 2 Report

Comments and Suggestions for Authors

It is an interesting topic.

It is an interesting case presentation, clearly presented in stages. The evaluation of the patient and the analysis of each stage led to the diagnosis. The histopathological examination, along with the morphological and immunohistochemical studies, were very important. The patient is a 9-year-old child, and it is good that the correct diagnosis was made.

I appreciate the work done for this study.

My comments are only intended to make the paper better. Good luck!

Author Response

Reviewer Comments:

Dear Reviewers,

Thank you for your thoughtful feedback and comments on our manuscript. We appreciate your efforts and the time you provided to improve the manuscript. Your insights have enhanced the quality of our study. Our point-by-point responses to your comments are shown in red font below. The changes to the manuscript are shown in red font.

Reviewer 2

It is an interesting topic.

It is an interesting case presentation, clearly presented in stages. The evaluation of the patient and the analysis of each stage led to the diagnosis. The histopathological examination, along with the morphological and immunohistochemical studies, were very important. The patient is a 9-year-old child, and it is good that the correct diagnosis was made.

I appreciate the work done for this study.

My comments are only intended to make the paper better. Good luck!

Response: Thank you for your positive feedback on our manuscript.

Reviewer 3 Report

Comments and Suggestions for Authors

Thank you for the privilege of reviewing the manuscript of the case report entitled: Diagnostic challenge of localized tenosynovial giant cell tumor in children. The article describes the case of a 9-year-old boy suffering from tenosynovial giant cell tumor (TSGCT) of the knee and highlights the frequently observed diagnostic challenges (for example related to the timely diagnosis of TSGCT).

The article is well presented and shows that TSGCT represents a diagnostic challenge and usually requires magnetic resonance imaging or histopathologic examination of a biopsy to establish the correct diagnosis of TSGCT.

Thank you for mentioning CSF-1R treatment in the Discussion section.

I think authors should highlight in the discussion section that CSF-1R inhibitors very likely will change the future treatment of TSGCT in children (I hypothesize that treatment with biologics will probably replace surgery as the Gold standard for treatment of primary or recurrent TSGCT in children).

I have some minor recommendations:

Please insert line numbers.

Page 2 line 2: replace “he sought” by “his family sought”.

Page 2 line 10 and entire manuscript: Provide units for CRP, WBC.

Page 3 line 19: replace “pyogenic arthritis can be considered as well.” by “septic arthritis must be ruled out.”.

Page 4 line 35: replace “resection, wither arthroscopic or open.” by “resection, either arthroscopic or open.”.

Author Response

Reviewer Comments:

Dear Reviewers,

Thank you for your thoughtful feedback and comments on our manuscript. We appreciate your efforts and the time you provided to improve the manuscript. Your insights have enhanced the quality of our study. Our point-by-point responses to your comments are shown in red font below. The changes to the manuscript are shown in red font.

Reviewer 3

Thank you for the privilege of reviewing the manuscript of the case report entitled: Diagnostic challenge of localized tenosynovial giant cell tumor in children. The article describes the case of a 9-year-old boy suffering from tenosynovial giant cell tumor (TSGCT) of the knee and highlights the frequently observed diagnostic challenges (for example related to the timely diagnosis of TSGCT).

The article is well presented and shows that TSGCT represents a diagnostic challenge and usually requires magnetic resonance imaging or histopathologic examination of a biopsy to establish the correct diagnosis of TSGCT.

Thank you for mentioning CSF-1R treatment in the Discussion section.

I think authors should highlight in the discussion section that CSF-1R inhibitors very likely will change the future treatment of TSGCT in children (I hypothesize that treatment with biologics will probably replace surgery as the Gold standard for treatment of primary or recurrent TSGCT in children).

I have some minor recommendations:

Please insert line numbers.

Response: Thank you for your positive feedback and comments on our manuscript. We have added line numbers.

Page 2 line 2: replace “he sought” by “his family sought”.

Response: The sentence was revised accordingly (line 31).

Page 2 line 10 and entire manuscript: Provide units for CRP, WBC.

Response: We included the units as suggested (lines 38,39 and Line 61).

Page 3 line 19: replace “pyogenic arthritis can be considered as well.” by “septic arthritis must be ruled out.”.

Response: The sentence was revised as suggested (lines 67-68).

Page 4 line 35: replace “resection, wither arthroscopic or open.” by “resection, either arthroscopic or open.”.

Response: This statement was revised to improve clarity (line 124).